# Volume-of-Interest Aware Deep Neural Networks for Rapid Chest CT-Based COVID-19 Patient Risk Assessment

**DOI:** 10.3390/ijerph18062842

**Published:** 2021-03-11

**Authors:** Anargyros Chatzitofis, Pierandrea Cancian, Vasileios Gkitsas, Alessandro Carlucci, Panagiotis Stalidis, Georgios Albanis, Antonis Karakottas, Theodoros Semertzidis, Petros Daras, Caterina Giannitto, Elena Casiraghi, Federica Mrakic Sposta, Giulia Vatteroni, Angela Ammirabile, Ludovica Lofino, Pasquala Ragucci, Maria Elena Laino, Antonio Voza, Antonio Desai, Maurizio Cecconi, Luca Balzarini, Arturo Chiti, Dimitrios Zarpalas, Victor Savevski

**Affiliations:** 1Centre for Research and Technology Hellas, Information Technologies Institute, 6th km Charilaou—Thermi, P.O. Box 60361, 57001 Thessaloniki, Greece; tofis@iti.gr (A.C.); gkitsasv@iti.gr (V.G.); stalidis@iti.gr (P.S.); galbanis@iti.gr (G.A.); ankarako@iti.gr (A.K.); theosem@iti.gr (T.S.); daras@iti.gr (P.D.); 2Humanitas AI Center, Humanitas Research Hospital, Via Alessandro Manzoni 56, 20089 Rozzano, Italy; pierandrea.cancian@humanitas.it (P.C.); alessandro.carlucci@humanitas.it (A.C.); mariaelena.laino@humanitas.it (M.E.L.); victor.savevski@humanitas.it (V.S.); 3Radiology Department, Humanitas Research Hospital, Via Alessandro Manzoni 56, 20089 Rozzano, Italy; caterina.giannitto@humanitas.it (C.G.); federica.mrakic_sposta@humanitas.it (F.M.S.); giulia.vatteroni@humanitas.it (G.V.); angela.ammirabile@humanitas.it (A.A.); ludovica.lofino@humanitas.it (L.L.); pasquala.ragucci@humanitas.it (P.R.); luca.balzarini@humanitas.it (L.B.); 4Dipartimento di Informatica/Computer Science Department “Giovanni degli Antoni”, Università degli Studi di Milano, Via Celoria 18, 20133 Milan, Italy; elena.casiraghi@unimi.it; 5Department of Biomedical Sciences, Humanitas University, Via Rita Levi Montalcini 4, 20090 Pieve Emanuele, Italy; antonio.desai@humanitas.it (A.D.); maurizio.cecconi@hunimed.eu (M.C.); artur.chiti@hunimed.eu (A.C.); 6Emergency Department, Humanitas Research Hospital, Via Alessandro Manzoni 56, 20089 Rozzano, Italy; antonio.voza@humanitas.it; 7Intensive Care Unit, Humanitas Research Hospital, Via Alessandro Manzoni 56, 20089 Rozzano, Italy; 8Humanitas Clinical and Research Center—IRCCS, Via Manzoni 56, 20089 Rozzano, Italy

**Keywords:** COVID-19, artificial intelligence, deep learning, CT-based diagnosis, patient risk assessment, infection quantification, patient stratification

## Abstract

Since December 2019, the world has been devastated by the Coronavirus Disease 2019 (COVID-19) pandemic. Emergency Departments have been experiencing situations of urgency where clinical experts, without long experience and mature means in the fight against COVID-19, have to rapidly decide the most proper patient treatment. In this context, we introduce an artificially intelligent tool for effective and efficient Computed Tomography (CT)-based risk assessment to improve treatment and patient care. In this paper, we introduce a data-driven approach built on top of volume-of-interest aware deep neural networks for automatic COVID-19 patient risk assessment (discharged, hospitalized, intensive care unit) based on lung infection quantization through segmentation and, subsequently, CT classification. We tackle the high and varying dimensionality of the CT input by detecting and analyzing only a sub-volume of the CT, the Volume-of-Interest (VoI). Differently from recent strategies that consider infected CT slices without requiring any spatial coherency between them, or use the whole lung volume by applying abrupt and lossy volume down-sampling, we assess only the “most infected volume” composed of slices at its original spatial resolution. To achieve the above, we create, present and publish a new labeled and annotated CT dataset with 626 CT samples from COVID-19 patients. The comparison against such strategies proves the effectiveness of our VoI-based approach. We achieve remarkable performance on patient risk assessment evaluated on balanced data by reaching 88.88%, 89.77%, 94.73% and 88.88% accuracy, sensitivity, specificity and F1-score, respectively.

## 1. Introduction

In December of 2019, the World Health Organization (WHO) China Country Office was informed of cases of an unknown respiratory disease detected in Wuhan City, Hubei Province of China [1]. The cause of this respiratory disease was the severe acute respiratory syndrome coronavirus-2 (SARS-CoV-2) virus, recently named Coronavirus Disease 2019 (COVID-19), which has become a global challenge since then [2,3,4]. According to the World Health Organization (https://www.who.int/emergencies/diseases/novel-coronavirus-2019 (accessed on 10 March 2021)), from February 2020 till now, 110 million COVID-19 cases have been confirmed while the world has suffered approximately 2.5 million losses. In Italy in particular, where this study partially took place and the patient data were gathered, the Italian National Institute of Statistics (https://www.istat.it (accessed on 10 March 2021)) (Istat) published a report (https://www.istat.it/it/files/2020/07/Rapp_Istat_Iss_9luglio.pdf (accessed on 10 March 2021)) in July 2020, stating that between February 20th and May 31st of 2020, the COVID-19 integrated surveillance system (https://www.epicentro.iss.it/en/coronavirus/sars-cov-2-integrated-surveillance (accessed on 10 March 2021)) registered 32,981 deaths; among them, 46% (15,133) of deaths took place before March 31st and 42% (13,777) in April, regarded as the peak period in Italy, while 12% (4014) in May. In June, deaths kept decreasing, while in July and August, the mortality rate returned back to levels similar to those of the preceding years. However, from the middle of September, especially in Lombardy area, COVID-19 deaths started increasing again (https://bit.ly/2Jzhjsc (accessed on 10 March 2021)) and, at the present time (December 2020), daily deaths are, on average, more than 500, and Intensive Care Units (ICU) of many Hospitals in Italy are near saturation.

In times of crisis, such as the current COVID-19 pandemic, rapid and efficient patient diagnosis and prognosis assessment would highly improve patient care and reduce mortality rate by eliminating the time intervals between Emergency Department (ED) arrival and hospitalization. For a quick and precise patient risk assessment, Computed Tomography (CT) has been described as an important diagnostic tool [5,6], given its capability of reducing RT-PCR false negative results [7] and its superior sensitivity compared to chest X-ray [8]. Indeed, CT quantification of pneumonia lesions can timely and non-invasively predict the progression to severe illness [9], providing a promising prognostic indicator for clinical management of COVID-19. Early identification of patients with increased organ damage risk can lead to rapid decision making and, consequently, to outcome improvement. However, given the time needed for an expert to analyze a CT and the limited availability of human resources, it is impossible to cover the massive CT analysis needs. On the other hand, the recent advancements of Artificial Intelligence (AI) applied to visual data and beyond, have shown remarkable results not only in medicine [10] but also in several other healthcare fields [11,12].

Despite the fact that, in most cases, a CT exam is immediately acquired once the patient presents COVID-19 symptoms and enters the ED, the radiologists make the final decision for the patient’s care only when the clinical and laboratory data is acquired. To this end, the existence of artificially intelligent tools for effective and efficient diagnosis could serve as a crucial medical expert assistant offering rapid patient risk predictions and suggestions. This could lead to better management of hospital wards and, therefore, to improved patient care by aiding clinicians in the early assessment of patient conditions, before the availability of the clinical and laboratory test results.

Indeed, since the pandemic started spreading, several techniques have been proposed approaching either COVID-19 diagnosis, or scoring of patients based on different types of risk [13]. Nevertheless, as evidenced in Section 2, despite the differences of the existing approaches with respect to the machine learning models, algorithms and architectures, all of them must deal with the difficulty of obtaining a proper optimization of the model parameters due to the varying CT dimensions (number of slices per CT), coupled with the limited number and cardinality of the publicly available training datasets.

To this end, the work reported in this paper documents a new data-driven approach for fast, AI-assisted, COVID patient detection and the criticality of the disease. Moreover, it presents COVID-19_CHDSET Dataset, a newly collected and annotated dataset from a reference hospital in Milan Italy during the first wave of COVID-19 in Europe. To this end, we exploit the data contained to our new COVID-19 CT annotated dataset and propose a novel CT-based stratification approach to rapidly assess the infection severity and then, the risk for COVID-19 patients. The statistical variety of the data collected from patients with varying characteristics (as discussed in Section 3), allows us to train our models more effectively due to the relationship between a dependent variable and a set of independent variables (statistical models). In detail, we introduce a two-stage data-driven approach to classify COVID-19 patients into three risk classes (moderate, severe, extreme), based on their risk to be, respectively, discharged, hospitalize, or sent to ICU short after CT examination. We firstly train a pixel-wise segmentation model on lung CT slices to detect the volume-of-interest (VoI), that is, the sub-volume of the lung containing the most infected slices, instead of the whole CT volume, which contains healthy or non-informative slices and whose elevated voxel-grid resolution would require an abrupt down-sampling. Secondly, based on the detected VoI, we train various 2D and 3D classifiers to predict the COVID-19 patient risks. This two-stage cascade concept (data of interest selection → classification) is not new in the medical imaging field [14,15,16]; however, to the best of our knowledge, our approach is the first that is driven by a spatially coherent data representation that leads to better and robust performance.

Summarizing, the main contributions of the present research work are reported in the following:The publication and use of COVID-19_CHDSET Dataset, a new CT dataset of COVID-19 infected patients from Milan, a region early and intensively involved in the pandemic. The CTs are labeled according the patient hospitalization outcome.The introduction of a novel, staged, data-driven technique of spatially coherent sub-sampling of CT volumes by detecting Volumes-of-Interest (VoI). This leads to lightweight but highly informative CT data that allow us to train state-of-the-art 2D and 3D classifiers more effectively than existing techniques.The exhaustive experimentation among various CT-based risk-prediction approaches and 2D/3D classifiers for COVID-19 patient stratification, to assess their performance.

The rest of the paper is structured as follows—Section 2 recalls related works designed for aiding clinicians in the fight against COVID-19 pandemic. Section 3 describes the dataset used for developing and testing the deep models. In Section 4, we present the approach we designed and developed to effectively address COVID-19 patient stratification exclusively on visual CT data. In Section 5, we present a comparative evaluation of our approach for various backbone models in comparison with recent state-of-the-art techniques. Finally in Section 6, we overview the main contributions of the present work and discuss its advantages and limitations as well as potential avenues for future works.

## 2. Related Work

In this section, we overview related works proposed for combating COVID-19, and we mainly focus on AI-driven approaches based on Machine Learning (ML) and Deep Convolutional Networks (DCN) for segmenting and classifying CT or CXR data (for an extensive review of methods AI-driven methods we refer the reader to [13,17]).

Unfortunately, though a great deal of research work has been devoted to the development of methods for automated COVID-19 diagnosis [18], severity scoring or prognosis prediction, and outcome prediction [8,13], we are still far from reaching a solution. That is due to the lack of a unified and anonymized, multi-device and multi-ethnicity, appropriately annotated and shareable datasets containing enough samples to ensure a robust model validation against the COVID-19 disease variability. Indeed, while several CT imaging databases have been made publicly available (https://github.com/HzFu/COVID19_imaging_AI_paper_list#technical_CT (accessed on 10 March 2021)), they either contain limited samples, or their labels are missing or not validated.

Given the greater sensitivity of CT examinations when compared to chest X-rays [19], deep learning models aimed at COVID-19 diagnosis have the potential for reaching higher prediction performance. This is the reason why in the past 11 months a great deal of research effort has been devoted to the development of ML methods working on chest CTs and mainly applying transfer, incremental learning techniques [20] to extend the already proposed methods for the automatic segmentation of the lung boundaries [21], or to develop lung lesions segmentation methods and their lesion type classification [22].

In more detail, given the amount of literature related to COVID-19 diagnosis from CTs, and their successful, effective, robust, potentially interpretable results [23], in the past three months some authors have started to investigate the problem of prognosis and outcome prediction, by developing systems that either compute a severity score from CTs or, similar to the technique proposed in this paper, predict a patient outcome (death/intubation/survival [8]) for COVID-19 infected patients.

Though some promising results have been already proposed, the problem of risk prediction from lung CTs is still open due to the documented complexity of such prediction [24] and to difficulties raised by processing 3D volumes. To cope with the aforementioned problem, different strategies have been proposed, which may be grouped into two classes, depending on the type of the input data. More precisely, the methods in the first class train 3D models which employ down-sampled lung volumes. More specifically, in [18], the authors present two models for COVID-19 classification, with the first model utilizing the entire lung volume and the latter one operating on samples of it. Additionally, in [24] the authors propose Prior Attention Residual Blocks (PARL) based on 3D convolutions, and build a neural network architecture for classifying CT scans as non-pneumonia, ILD and COVID-19. A notable technique exploiting a 3D strategy [25] concatenated 3D deep learning models among which a 3D-inflated Inception architecture [26,27], to compute a severity score by learning from CO-RADS scores [28] assessed by experts. The complexity of the developed systems however required abrupt down-sampling and a training sample with a large cardinality, which is often not available.

Such techniques are opposed to 2D strategies, which avoid any lossy down-sampling by reformulating the 3D problem as a 2D prediction, where many per-slice classifications are computed and then aggregated to compute the final prediction. More specifically, in [29] CT slices are selected based on parenchymal abnormality and classification of COVID-19 positivity is estimated by fusing 2D CT slices with non-clinical data. Moreover in [30], they employ a shared-weight convolutional neural network (CNN) architecture that utilizes a series of CT slices to finally aggregate the features via a pooling operation to classify no pneumonia, community acquired pneumonia and COVID-19.

Completely different, and more stable (especially when limited training datasets are available) radiomics approaches are those proposed in [16,31]. Both papers exploit 2D CNN based architectures to convert the 3D classification problem into a 2D one, and therefore proceed by initially analyzing each 2D slice to segment the lesions. Then patient outcome is predicted by highly explainable classifiers such as Support Vector Machines [32,33], or Random Forests [34,35], pooling the results of the 2D analysis with the knowledge carried by clinical data. Additionally, in [36], a similar approach is followed, by utilizing CNNs as well as a swarm-based feature selection algorithm (Marine Predators Algorithm) to classify COVID from non-COVID frontal cardiothorasic X-rays. They use the Inception CNN architecture [27] to extract features from CXR images and use the Marine Predators Algorithm to select the relevant features from the X-rays. Another very recent approach proposed in [37], involves more traditional methods comprising feature extraction and selection via an entropy-based fitness optimizer, feature fusion to finally classify CT images as COVID-19 and healthy by utilising a Naive Bayes Classifier. Additionally, in [38], the authors employ Random Forests to classify whether a patient needs ICU admission via a holistic approach, that is, by employing CT as well as demographic and other related metadata for training.

Among the different 3D to 2D approaches we surveyed, the method proposed by [39] is based on the understanding that clinicians assess the COVID-19 infection severity by observing both the type and the extent of the COVID-19 patterns (lesions) visible in the CT. Therefore authors use an effective 2D segmentation network, called Inf-Net (the code of Inf-Net is available at https://github.com/DengPingFan/Inf-Net (accessed on 10 March 2021)), which has been adapted to the slices used in this work (see Section 4.1), to extract COVID-19 lesions from 2D slices, and leave the final decision to experts.

Though the problem of COVID-19 diagnosis, prognosis and outcome prediction is still an open problem, one of the main strengths of almost all the approaches is the interpretation of the computed predictions through techniques such as grad-CAM [40], which allows us to show the areas that mostly contributed to the slice deep feature computation. Indeed a positive, nowadays established trend in the field of machine learning, regards the need of computing explanations able to motivate the computed predictions; this is especially necessary in medical applications where effective diagnosis and prognosis predictions may improve the patients’ survival rate.

According to previous literature [16,39,41], the effectiveness of deep models for COVID-19 diagnosis from CTs may be improved by attention-based strategies allowing a preliminary extraction of most informative sub-volumes, used for patient classification. By focusing on the most infected slices, the model complexity is reduced, so that abrupt and lossy image reduction can be avoided. Further, the classifier performance is increased when transfer learning techniques are applied, which allow dealing with the limited cardinality of the available training sets.

## 3. COVID-19_CHDSET Dataset

For our study, we created and used a new publicly available dataset (the anonymized and annotated dataset is available upon request and approval. Please send your request to biblioteca@humanitas.it with subject [COVID-19_CHDSET] and cite the title of the article, your name, affiliation and purpose of the request, or contact victor.savevski@humanitas.it for further details. The request will be processed and additional information may be required.) including 626 chest CTs from 497 COVID-19 patients (F/M: 171/326, median age: 67±14.53 years, age range: [27, 94]), with the 76% of them presenting at least one comorbidity. These patients entered ED between March and April of 2020 and underwent chest CT imaging with RT-PCR proven COVID-19 infection. For some of them, beyond the CT acquired at their admission, extra CTs were eventually acquired in the subsequent days.

The CTs were acquired through 3 different scanning machines (Ingenuity CORE^TM^ and Ingenuity^TM^ CT from Philips, and Revolution EVO^TM^ from GE Medical Systems), with a mean horizontal and vertical slice resolution of 0.785±0.087 cm and an average inter-slice thickness of 2.23±0.554 cm (range [0.625, 5], median 2. With these settings, we extracted CT scans in DICOM format, having, on average 157±38 slices (range [28, 351]), and, for the 90% of the CTs, a 2D slice size of 512×512 pixels (the remaining 10% had slice size ranging from 452×452 pixels to 768×768 pixels).

Each CT image was labeled according to the patient’s risk of severe complications, which was defined following the patient’s type of hospitalization or eventual discharge after the CT (the class labeling was conducted exclusively from the actual hospitalization path each patient underwent, meaning that there is no “clinical” pattern for each risk level since patients are assessed based on lung infection rate, age, clinical factors and other comorbidities.). In particular, 118 CTs were labeled as moderate risk (Class 0) since the patient was discharged after CT execution, 468 CTs were labeled as severe risk (Class 1) since the patient was hospitalized after CT execution, but was not sent to the ICU in the following 12 h, and 40 CTs were labeled as extreme risk (Class 2) since CT scans were executed on patients already in the ICU, or admitted to the ICU within the following 12 h.

On top of that, the CT slices were processed by experienced radiologists who manually drew pixel-wise COVID-19 lesion annotations. In detail, the radiologists annotated one-by-one the images of the CTs by drawing 2D contours around the identified lesions, as shown in Figure 1. The annotated lesions, according to the RADLEX lexicon [30], include *Consolidation* [Radlex IDentifier (RID): 43255], *Crazy Paving Pattern* [RID: 43256], *Ground Glass Opacity* [RID: 28531], *Vascular Dilatation* [RID: 4743], *Subpleural Bands & Architectural Distortion* [RID: 34261] and *Traction Bronchiectasis* [RID: 28528].

For further details on our dataset and the way we used it in the present work, see Section 5.1.1 and Table 1.

## 4. COVID-19 CT-Based Patient Risk Assessment

Volumetric imaging problems often rely on 3D convolutional neural networks; nevertheless, existing hardware (e.g., GPU memory) cannot afford the processing of high-resolution CT data. This necessitates the use of lower-resolution volumes (e.g., abruptly down-scaled volumes) leading to lower diagnostic accuracy since the lesions and other pathological patterns are blurred, or totally canceled when having small extent. On the other hand, the direct use of 3D data allows deep models to solve 3D problems by taking into account the spatial and structural relationships between neighboring pixels in volumetric neighborhoods. Given the aforementioned considerations, we designed the processing pipeline described in this work in order to analyze the CT data in a way that will trade-off the spatial resolution size and the amount of processed data. More precisely, by emulating the visual analysis performed by human experts,’ we firstly identify and extract the most infected lung 3D segments (see Section 4.1), which we regard as the most appropriate to indicate the COVID-19 severity level of the patients. In a subsequent stage, as described in Section 4.2, the detected VoIs are fed to deep classifiers to predict the COVID-19 patient risks class from *moderate (Class 0)*, to *severe (Class 1)*, and *extreme risk (Class 2)*. The overall concept of our approach is illustrated in Figure 2, while the processing pipeline is schematized in Figure 3.

### 4.1. COVID-19 Quantification for VoI Detection

We begin with the consideration that the most infected lung 3D segment is the most informative and indicative with respect to patient’s lung severity level and, therefore, ideal to drive a deep classifier to successfully predict COVID-19 patient’s risk level. To this end, we train our first-stage model to segment COVID-19 patterns (infected areas) allowing us to quantify the infection per slice across the CT depth. We detect the most infected 3D segment, termed VoI, by extracting $d^{\prime }<d$ from the *d* slices of the CT around the slice identified as the most infected, that is, with the largest total infected area in terms of COVID-19 labeled pixels.

To segment the infected areas from each slice, we based our model upon Semi-Inf-Net network described in [39] (see Figure 3), predicting a binary mask for the COVID-19 infected areas. This network prevents the problem of diminishing gradients, by attending to all of its five convolutional layers. Indeed, an *Edge Attention* module, linked to the second convolutional layer, allows guiding the deep features resulting at that stage by an edge map; a *Parallel Partial Decoder* (PPD) module produces a coarse infection map by merging the results from the third, fourth and fifth layer, and therefore allows supervision of their multi-scale, concurrent job; the feature map resulting from the PPD guides the work of three *Reverse Attention* modules (RAs), each linked to one of the last convolutional layers to produce a lesion map by merging the results of the PPD, the activation map from their directly connected layer, and that resulting from the work of the first two convolutional layers. The multiple maps, computed by the three RAs and PPD, are aggregated and fed to a *Sigmoid* activation layer which outputs a binary mask of the infected areas in the input slice. Thanks to the multi-scale analysis produced by the attention-based strategy, EA, RAs, and PPD concurrently operate to improve the lesion localization and its segmentation. Further, since all modules (EA, RAs, PPD, and output) are supervised by retro-propagating the cross-entropy loss between the segmentation map and the manual segmentation produced by experts, the network overcomes vanishing gradients [42].

Once the COVID-19 mask of a CT slice is segmented, the infection level is computed as the sum of the labeled pixels. To this end, we quantify the infection of all slices of a CT and, subsequently, the VoI is constructed by including the $d^{\prime }$ slices around *I*, where *I* denotes the most infected one.

### 4.2. VoI Classification for COVID-19 Patient Risk Assessment

We subsequently apply deep classifiers on the detected VoI to assess the COVID-19 patient’s risk. One of the major contributions of VoI detection is that it enables experimentation with various neural network architectures, including 3D ones due to the data spatial coherency. To this end, we experiment with various 2D and 3D state-of-the-art models to explore their performance with regards to patient risk assessment based on CT data.

With respect to 2D models, our strategy is to approach the risk assessment by classifying the detected VoIs in a per-slice manner. Among the various 2D backbone models we explored, ResNet-101 [43] and DenseNet-201 [44] showcased the best performance. We trained these models to label each slice separately, while the concluding VoI classification is obtained with the use of majority voting. On the other side, regarding the 3D models, we experimented with ResNet3D [45], MixedConv [46] with its initial 3D layers specifically aimed at extracting volumetric information to be further processed by 2D layers in the top abstract levels, and ResNet2Dplus1D [46], a pseudo-3D model based on 2D convolutions on the horizontal planes and parallel 1D convolutions along the vertical axis.

## 5. Evaluation

In this section, we present the experimental methodology (Section 5.1) we followed to evaluate the performance of the proposed approach and the respective CT-based only patient risk assessment results (Section 5.2). The implementation details for the execution of this evaluation are documented in Appendix A.

### 5.1. Experimental Setup

Given the COVID-19_CHDSET dataset described in Section 3, we present the training/testing sets we used to train and assess our models in Section 5.1.1. In Section 5.1.2, we present the strategies we designed to assess our approach against other strategies, while in Section 5.1.3, we discuss the metrics we used in these experiments. On top of that, aiming to provide the readers with a practical and real-life reference, experienced radiologists participated in this study and assessed the same CTs following the assessment protocol described in Section 5.1.4.

#### 5.1.1. COVID-19_CHDSET Dataset Splits

In this work, COVID-19_CHDSET usage is twofold; while the binary lesion annotations are used as ground truth to allow pixel-wise training and testing of the segmentation model (see Section 4.1), the 3-class labels allow for training and testing the risk assessment classifiers (see Section 4.2).

Regarding the former, 2550 and 320 CT slices were used to train and test the segmentation model for lesion detection. For the latter, we randomly split into two non-intersecting stratified training and test sets composed of approximately 80% and the 20% of all available CT samples, respectively. To handle the large class imbalance, a new dataset (COVID-19_CHDSET_OS_) was obtained by over-sampling the under-represented classes in the training set, which was therefore balanced [47]. In particular, Class *moderate risk* was oversampled with a ratio of 7/2, while Class *extreme risk* was over sampled with a ratio of 11/1. In this way, the re-balanced training set had 332 moderate risk samples, 374 samples from patients at severe risk, and 341 samples from patients at extreme risk.

The 3-class patient risk distribution as well as the distribution in the over-sampled splits are illustrated in Table 1. Note that, to allow a comparison to radiologists’ assessment, we selected an under-sampled balanced dataset, from which the 20% of CTs (which accounted for 27 CTs) were randomly sampled to compose a balanced test set. The low number of test images is due to experts’ shortage of time; however, this experiment gave us the chance to test the robustness of the proposed methods with respect to a training dataset which, though balanced, has a limited cardinality.

In practice, the whole dataset is reduced in size so that the cardinality of each class is diminished to the cardinality of the less represented class. From this dataset, we randomly extracted the 80% of samples for training and the remaining samples for testing. The class distribution for this dataset (COVID-19_CHDSET_US_) is shown in Table 1. The predictions on the 27 images in the test set of COVID-19_CHDSET_US_ were compared to those of radiologists. The dataset details are publicly available (https://vcl.iti.gr/COVID/ (accessed on 10 March 2021)) to allow the reproducibility of the results and further research in the field.

#### 5.1.2. CT-Based Risk Assessment Strategies

Complementary to the above human- versus machine-prediction comparison, we conducted experiments to compare our approach against different CT-based risk-assessment strategies concerning computational load and performance. Actually, we evaluate two recent strategies against VoI to essentially reduce the number of input voxels/pixels and predict the patient risk levels:SSoI: is applied in a similar fashion to VoI. Precisely, to avoid any lossy volume down-sampling, by essentially reducing the height of the treated volume, we experimented with an alternative approach based on recent works [16,39]. In detail, CT slices are analyzed to quantify the infection, and the most infected ones (not necessarily consecutive) are considered to compose a stack of slices-of-interest (SSoI); Volume: the second and mostly common approach we experimented with, considers the whole volume and down-scales it to reach manageable data sizes and computational time.

#### 5.1.3. Evaluation Metrics

Regarding the evaluation metrics, we use *Dice* and *Intersection-Over-Union* (*IoU*) scores to assess the similarity between the predicted COVID-19 segmentation masks and ground-truth as provided by expert radiologists. The mean over all the Dice and mIoU scores computed for all slices in the test set were used as global performance measures. For the assessment of VoI classification models, we computed the macro-average multi-class *accuracy*, *sensitivity*, *specificity*, *AUC*, and *F1-score* over the test sets.

#### 5.1.4. Radiologists’ CT-Based Risk Assessment Protocol

To complement our study and offer a comparative reference to the readers concerning the difficulty level of this task, three experienced (with at least five years of experience) radiologists assessed COVID-19_CHDSET_US_. Based on previously acquired knowledge, they evaluated each CT starting from left and right lobe analysis and then extended to the mediastinum section anomalies check. Following this protocol, they classified each CT in the test set of COVID-19_CHDSET_US_ as *moderate*, *severe*, and *extreme* risk for the patient. The evaluation was performed within a blind setup: each radiologist evaluated the CT without contact or information regarding predictions from other radiologists or data-driven models.

### 5.2. Results

In this section, we firstly present the per-slice lesion segmentation results (Section 5.2.1) for COVID-19 quantification and VoI detection and, subsequently, we report and discuss the comparative evaluation of the risk assessment outcomes achieved by our different risk prediction models against similar recent techniques and experts’ predictions (Section 5.2.2).

#### 5.2.1. Lesion Segmentation Results

In Figure 4, we depict qualitative results of the proposed segmentation model and ground-truth COVID-19 lesions, along with *mDice* and *mIoU* quantitative results on the 320 testing slices. Even though many samples are often rotated, translated, and sometimes cut, the model is able to reach high performance, witnessing the similarity between the computed segmentation masks and the manual annotations provided by the experts.

Besides, the algorithm is able to detect the infections even in cases where the infected area is quite limited and/or distributed among the lungs. Given the high scores in the COVID-19 segmentation task, as shown in Table 2, we consider the segmentation model a reliable means to quantify and assess the lung infection level.

Using this initialization, and given a training set extracted among the slice for which experts provided a contour (see Section 5.1.1 for a description of the training and test sets) the model was trained for 100 epochs with batch size of 2, and by using Stochastic Gradient Descent, with an initial learning rate of 0.001, and momentum 0.7.

#### 5.2.2. CT-Based Risk Assessment Results

We aim to highlight the effectiveness of the presented classifiers in conjunction with the proposed VoI detection against most similar techniques for chest CT volume attention and data pre-processing, i.e., techniques working on *SSoI* [16,39]. Since we work on a per slice basis, for this comparison we tested only the models that do not fully exploit a volumetric representation (ResNet2DPlus1D and DenseNet201).

The number $d^{\prime }$ of slices composing the VoI and the SSoI was experimentally chosen in the range of [4, …, 16] aiming to maximize the multi-class F1-score on the training set. In particular, to allow a coherent comparative evaluation of the different architectures, we considered $d^{\prime }=10$ since, on the average of all the models, this value allows obtaining the highest mean F1-score.

The results computed by the different models on the COVID-19_CHDSET_OS_ test set are reported in Table 3, while the comparison between radiologists performance and machine performance, performed on the COVID-19_CHDSET_OS_ dataset is reported in Table 4. Figure 5 and Figure 6, the ROC are showing allowing a visual comparison of the ROCs achieved on, respectively, the COVID-19_CHDSET_OS_ and the COVID-19_CHDSET_US_ datasets by DenseNet201-VoI and DenseNet201-SSoI for the three classes.

Our outperforming model, DenseNet201-VoI favorably compares to DenseNet201-SSoI by showing higher and comparable results in COVID-19_CHDSET_OS_ and COVID-19_CHDSET_US_, respectively. In detail, DenseNet201-VoI reaches 88.88%, 89.77%, 94.73% and 88.88% accuracy, sensitivity, specificity and F1-score, respectively, on COVID-19_ CHDSET_US_ test set, and 81.42%, 84.45%, 87.32% and 78.82% accuracy, sensitivity, specificity and F1-score on COVID-19_CHDSET_OS_ test set. The lower performance achieved by SSoI models when compared to VoI models, suggests that maintaining the spatial coherency of the data allows increasing performance. This is especially true when using (pseudo-)3D models, as suggested by the dramatically low performance of the pseudo 3D model ResNet2DPlus1D-SSoI.

Furthermore, observing the radiologists’ results on the COVID-19_CHDSET_US_ dataset, it might be noted that, even though deep models achieve high scores in patient risk assessment based only on CT images, radiologists struggle on this task, achieving lower results in all metrics as shown in Table 4. That is because models have been explicitly trained on the task, while experts are not familiar with this procedure. In practice, medical experts assess patients’ condition and risks based on additional significant information from clinical and biochemical data.

Observing the Receiver Operating Curves (ROCs) and the respective Areas Under the Curve (AUCs) in Figure 5 and Figure 6 for COVID-19_CHDSET_OS_ and COVID-19_CHDSET_US_ datasets, respectively, we observe that the curves for the *severe* and *extreme risk* classes, exhibit desired ROC properties in conjunction with high AUC values. In contrast, the ROC curve of the SSoI classifier displays less accurate predictions except for the extreme risk class that seems to be easier to distinguish among the three classes. Such ROCs highlight that the networks working on the VoI are more tolerant concerning lower specificity as compared to lower sensitivity since a type II error (false negative) could be disastrous in a medical setting. To further support our results, we provide a hypothesis test following the approach [48] for each 2D and 3D classifier. Our purpose is to showcase that the performance of our best model (i.e., DenseNet201-VoI) is significantly better than the best 3D approach (i.e., ResNet2DPlus1D-VoI) and the radiologists performance in the COVID-19_CHDSET_US_ test set. Particularly, as proposed in [49] we frame our hypothesis as follows:$$H_{0}:p_{1}=p_{2}$$
$$H_{0}:p_{1}<p_{2}$$
where $p_{2}$ is the global accuracy of DenseNet201-VoI and $p_{1}$ the accuracy of ResNet2DPlus1D-VoI, and radiologists accordingly. Then the rejection region is given by $Z<z_{a}$, where $z_{a}$ = 1.645 for 5% level of significance and the test statistic *Z* is calculated by the following equation $Z=\left(p_{1}-p_{2}\right)/\sqrt{(}2\times p\times \left(1-p\right))/n$. The calculated test statistic value for ResNet2DPlus1D-VoI and radiologists is −2.13 and −3.63 accordingly, and as such we can state that the performance of our classifier is better than the compared ones with 95% confidence level.

**Ablation Study.** To further showcase the effectiveness of our VoI-based approach, we conducted two extra experiments to ablate our contributions. At first, to assess the importance of VoI detection, instead of analyzing the sub-volume centered on the most infected slice, we performed experiments by using as input a sub-volume centered on a slice randomly (RVoI) selected among those segmented as infected by the segmentation model (*ResNet3D-RVoI*, *MixedConv-RVoI*, *ResNet-101-RVoI*, *ResNet-101-RVoI*, *DenseNet-201-RVoI*). Secondly, we totally removed the first VoI/SSoI detection stage, working on the whole, opportunely reduced, volume. To achieve that, we reduced the volume to 128 × 128 × 100 voxel-grid resolution since this allowed obtaining a number of input voxels comparable to those of models working on the VoI or SSoI (512 × 512 × 10).

Results shown in Table 5 and Table 6 highlight that DenseNet201-VoI outperforms the rest of the models, showing the remarkable performance of the 3D-to-2D decomposition against 3D strategies in this task. The lower performance of all models on COVID-19_CHDSET_US_ is rather due to the much lower cardinality of the available training set. It is highlighted that the models working on VoI always achieve a higher F1-score than their counterparts, working on the whole volume. For this last sentence I would put a bar chart that allows a direct visual comparison.

To allow clinicians supervise the automated risk assessment task, the system computes explanations in forms of activation maps. In particular, the Guided gradient-weighted Class Activation Mapping (Guided Grad-CAM [40]) is exploited to visualize the model activation maps on the lesion areas and pathological patterns, permitting model explainability to medical experts. In detail, the class-discriminative localization maps use the feature maps produced by the last convolutional layer of our CNN to assign a score for a specific class. As per-experts assessment, the activation maps accurately localized infected areas. The class activation maps computed for DenseNet201-VoI on different slices are shown in Figure 7.

For 2D classifiers, all images were resized to 330×330. Our models were initialized with pre-trained weights on ImageNet [50] and trained for 120 epochs with a batch size of 4. We trained the model with Stochastic Gradient Descent (SGD) [51], an initial learning rate of 0.001 and a momentum of 0.7. Both focal loss [52] and categorical entropy loss were used, with the best results obtained by the latter. ResNet101 and DenseNet201 consist of 40 M and 20 M parameters, respectively.

All the 3D models were initialized with random weights, used cross-entropy loss, and the Adam [53] optimizer. Initial learning rate was set to 0.001, which was decreased by a factor of three each time the training accuracy was not improved for more than 15 epochs. The models were trained for 200 epochs with the maximum attainable batch size for our GPU (NVIDIA Tesla K80). ResNet3D, MixedConv and ResNet2Plus1 consist of 64 M, 46.6 M and 45.8 M parameters, respectively.

The rest of the parameters that are mentioned were set to the default values. The implementation of all the 2D and 3D models was performed using the PyTorch deep learning library [54]. Further implementation details, code and dataset usage instructions to re-produce the presented results can be found at https://vcl.iti.gr/COVID/ (accessed on 10 March 2021).

## 6. Discussion and Conclusions

Compromised lung volume is among the most accurate COVID-19 patient outcome predictors [55], so that CT investigation is often requested as the imaging modality together with biochemical tests when patients with serious COVID-19 symptoms access the EDs. Since the results of biochemical tests are available hours later, an effective patient’s risk assessment from CT volumes is desirable.

Given the already proven effectiveness of medical AI/ML solutions, especially in vision-based problems, we consider that a data-driven approach operating on CT visual only data will facilitate and boost risk assessment of the COVID-19 patients. To this end, we introduce a new, two-stage method to assess the severeness of COVID-19 patients’ exclusively based on chest CT volumes. Specifically, exploiting a new CT dataset, COVID-19_CHDSET, where COVID-19 lesions were manually annotated by experts, and considering the impractical computational loads affecting neural models working on CT at their original resolution, we propose to avoid any lossy volume down-sampling by driving our models to focus on the most infected segments of the CT volume, detected and extracted with the use of a COVID-19 lesion segmentation model.

Observing the outcomes of this study, we realize that data-driven models achieve better performance when processing consecutive slices centered on the mostly infected sub-volume (VoI) of the CT. Specifically, focusing on our best performing model, i.e., DenseNet201-VoI, we reach 88.88%, 89.77%, 94.73% and 88.88% accuracy, sensitivity, specificity and F1-score, respectively, on COVID-19_CHDSET_US_ test set, and 81.42%, 84.45%, 87.32% and 78.82% accuracy, sensitivity, specificity and F1-score on COVID-19_CHDSET_OS_ test sets. These results are considered so remarkable that they enable the deployment and use of these models in the HUMANITAS hospital in pilot phase, which was the initial goal of this work. Of course, given the relatively limited number of samples in comparison with COVID-19 cases, during this phase, all potential limitations of the models, such as bias to the training data, will be tracked to show the weaknesses and potential threats-to-validity of the experimental results.

Contrary to our expectations, 3D models providing one-shot decision on VoI (i.e., ResNet3D, MixedConv, ResNet2Plus1D), show lower performance when compared to models decomposing the 3D data into 2D per-slice predictions (through ResNet-101 or DenseNet-201) aggregated with majority voting. This was not expected given the capability of the 3D models to learn spatially coherent features due to the 3D input and the fact that the number of model parameters are not varying enough to explain it. This unexpected finding constitutes one of the current problems to solve and task under research for our team.

On top of that, we are going to further exploit COVID-19_CHDSET by exploring novel computer vision approaches that will allow medical decision making on many other aspects with respect to the health condition of the patients. The multi-lesion annotations can be fully exploited along with clinical data, leading to better performance as well as multi-tasking models (disease classification—severity—infected areas, etc.). We will further investigate augmentation and disentanglement techniques in order to overcome model overfitting and achieve inference generalization, training models that can be easily utilized by and deployed to other hospitals and clinics showing high effectiveness.

In conclusion, we can assume that the proposed AI models may be a valuable tool for aiding clinicians in the early assessment of patient conditions and prognosis forecast in the ED, when laboratory test results are not yet available. CT exams are usually acquired as soon as the patient presents COVID-19 symptoms, however evaluation is performed by radiologists only when all the clinical and laboratory information is acquired to provide a proper patient condition evaluation. To this end, it is crucial to save time to quarantine positive patients to prevent infection spread across a hospital’s units. 

## Figures and Tables

**Figure 1 ijerph-18-02842-f001:**
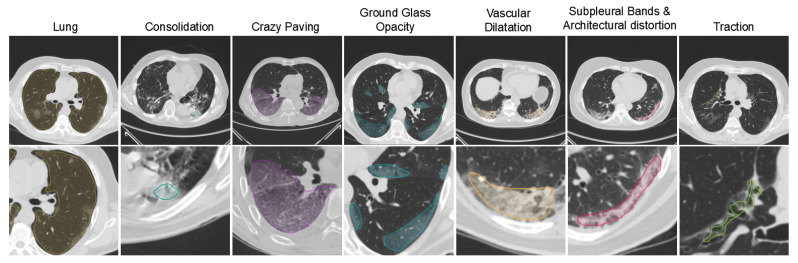
The various lesion annotations are illustrated per column. The second row gives in detail the the respective annotations.

**Figure 2 ijerph-18-02842-f002:**
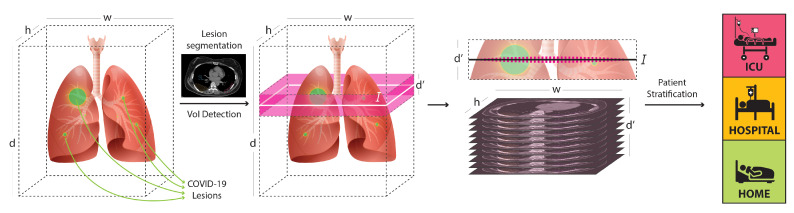
For a CT scan with size w×h×d, where *w*, *h* and *d* are the slice-width, slice-height, and number of slices, respectively, the most informative volume of interest (VoI) is composed by $d^{\prime }$ adjacent slices around *I*, with *I* being the most infected slice, that is, the “center of infection.” Discarding the less informative slices, we process the VoI by retaining its detailed information, without any abrupt down-sampling. To this end, the expressive power of the models is focused on the analysis of all the detailed information carried by the salient, informative slices.

**Figure 3 ijerph-18-02842-f003:**
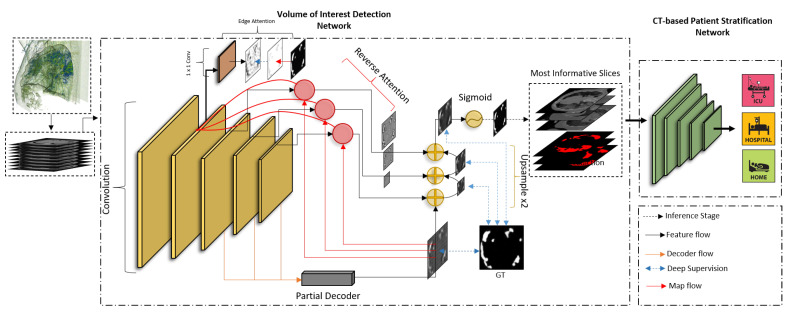
Overview of the suggested two-stage approach composed of our volume-of-interest (VoI) detection through lesion segmentation to quantify Coronavirus Disease 2019 (COVID-19) infection on the various 3D segments of the lung CT, and patient stratification through VoI classification models.

**Figure 4 ijerph-18-02842-f004:**
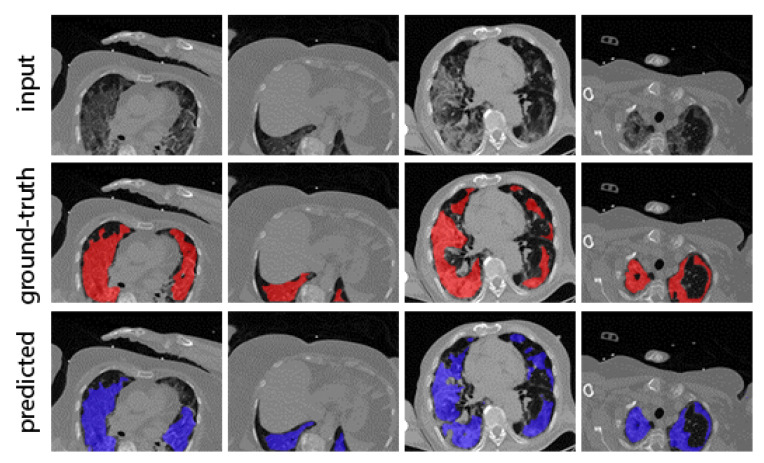
Qualitative and quantitative results of the proposed segmentation model. Input CT slices, ground truth (**red**) and predicted (**blue**) infected segments are shown in the first, second and third column, respectively.

**Figure 5 ijerph-18-02842-f005:**
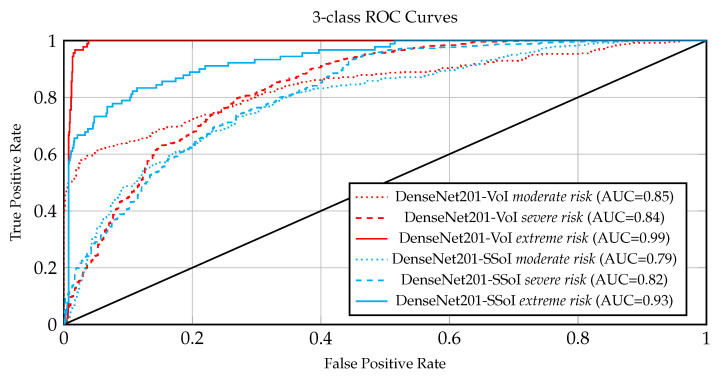
ROCs of DenseNet201-VoI and DenseNet201-SSoI models on COVID-19_CHDSET_OS_ with one ROC curve plotted for each class.

**Figure 6 ijerph-18-02842-f006:**
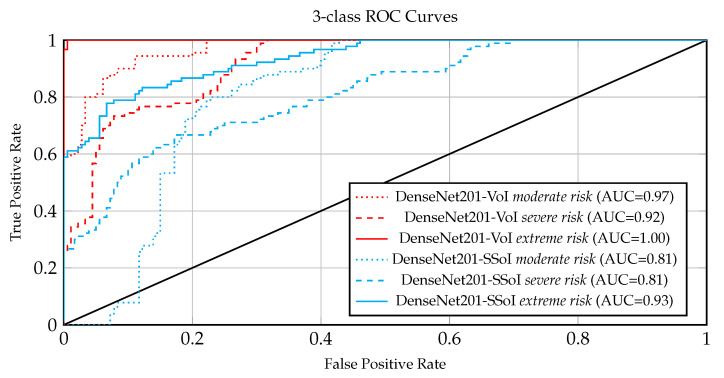
Receiver operating curve (ROC) of DenseNet201-VoI and DenseNet201-SSoI models on COVID-19_CHDSET_US_ with one ROC curve plotted for each class.

**Figure 7 ijerph-18-02842-f007:**
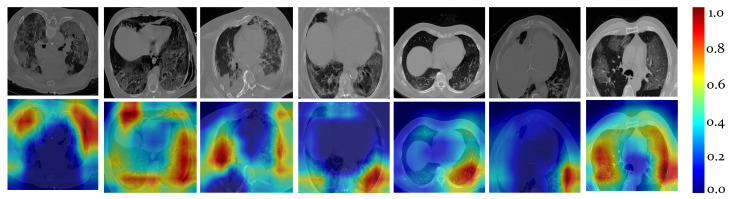
Visualizations of the Gradient Weighted Class Activation map (Grad-CAM) for chest CT slices from seven COVID-19 patients from the test sets. The first four columns correspond to extreme risk while the latter corresponds to severe risk classes. Red and blue regions correspond to maximum and minimum scores for the given class respectively, thus, red to yellow colors represent the most discriminative regions of interest in the corresponding CT images for patient risk assessment. For simplicity, we normalize the scores (i.e., [0,1]).

**Table 1 ijerph-18-02842-t001:** Distribution of the available Computed Tomography (CT) scans into the three classes, the training, and the test set.

COVID-19_CHDSET	Training Set	Testing Set	Total
moderate risk	94	24	118
severe risk	374	94	468
extreme risk	31	9	40
**Total**	499	127	626
**COVID-19_CHDSET_OS_**	**Training Set**	**Testing Set**	**Total**
moderate risk	332	24	356
severe risk	374	94	468
extreme risk	341	9	350
**COVID-19_CHDSET_US_**	**Training Set**	**Testing Set**	**Total**
moderate risk	31	9	40
severe risk	31	9	40
extreme risk	31	9	40

**Table 2 ijerph-18-02842-t002:** Results on lesion segmentation. Mean dice, IoU and their 95% confidence limits.

	mDice (%)	mIoU (%)
Lesion Seg.	97.8 (+1.1–7.9%)	95.6 (+1.5–8.7%)

**Table 3 ijerph-18-02842-t003:** Results on the COVID-19_CHDSET_OS_ test set. Best results are highlighted in **bold**.

		Acc (%) ↑	Sens (%) ↑	Spec (%) ↑	F1 (%) ↑
3D	ResNet2Plus1D-VoI	79.58	70.29	83.67	70.02
	ResNet2Plus1D-SSoI	74.01	33.33	66.67	28.35
2D	DenseNet201-VoI	81.42	**84.45**	87.32	78.82
	DenseNet201-SSoI	**82.85**	81.37	**87.54**	**79.15**

**Table 4 ijerph-18-02842-t004:** Results on the COVID-19_CHDSET_US_ test set. Best results are highlighted in **bold**.

	Model	Acc (%) ↑	Sens (%) ↑	Spec (%) ↑	F1 (%) ↑
3D	ResNet2Plus1D-VoI	62.90	62.90	81.4	63.06
	ResNet2Plus1D-SSoI	33.33	33.33	66.66	16.66
2D	DenseNet201-VoI	**88.88**	**89.77**	**94.73**	**88.88**
	DenseNet201-SSoI	82.33	82.57	90.89	82.33
	Radiologists	40.74	39.19	70.37	39.05

**Table 5 ijerph-18-02842-t005:** Ablation results on HUMC19-CT_US_ test set. Best results are highlighted with **bold**.

	Model	Acc (%) ↑	Sens (%) ↑	Spec (%) ↑	F1 (%) ↑
3D	ResNet3D-VoI	66.67	66.67	83.31	65.85
	ResNet3D-RVoI	72.54	51.97	77.58	52.32
	MixedConv-VoI	51.85	51.85	75.92	45.71
	MixedConv-RVoI	66.2	46.35	73.10	45.78
	ResNet2Plus1D-VoI	62.90	62.90	81.4	63.06
	ResNet2Plus1D-RVoI	42.85	33.33	66.66	20.00
	ResNet2Plus1D-Volume	33.33	33.33	66.66	16.66
2D	ResNet101-RVoI	76.42	83.80	86.35	76.42
	ResNet101-VoI	85.18	85.60	92.32	85.60
	DenseNet201-RVoI	77.77	80.68	89.47	78.88
	DenseNet201-VoI	**88.88**	**89.77**	**94.73**	**88.88**
	DenseNet201-Volume	72.14	72.68	79.58	61.33

**Table 6 ijerph-18-02842-t006:** Ablation results on the HUMC19-CT_OS_ test set. Best results are highlighted with **bold**.

		Acc (%) ↑	Sens (%) ↑	Spec (%) ↑	F1 (%) ↑
3D	ResNet3D-VoI	80.99	71.14	82.93	71.11
	ResNet3D-RVoI	68.31	52.32	73.69	52.32
	MixedConv-VoI	76.76	67.31	82.07	65.85
	MixedConv-RVoI	67.61	57.47	75.64	55.55
	ResNet2Plus1D-VoI	79.58	70.29	83.67	70.02
	ResNet2Plus1D-RVoI	71.83	42.11	73.71	41.82
	ResNet2Plus1D-Volume	68.84	33.33	66.67	27.18
2D	ResNet101-RVoI	76.00	66.30	77.44	69.86
	ResNet101-VoI	76.42	83.80	86.35	77.27
	DenseNet201-RVoI	74.29	74.27	81.80	74.29
	DenseNet201-VoI	**81.42**	**84.45**	**87.32**	**78.82**
	DenseNet201-Volume	68.10	53.07	69.98	46.45

## Data Availability

Data is only available on request due to privacy restriction. The data presented in this study are available upon request to Humanitas Research Hospital. Please contact Dr. Caterina Giannitto at caterina.giannitto@humanitas.it to proceed with the data access request. The data are not publicly available due to privacy agreement limitations.

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
