# Peer review of "Volume-of-Interest Aware Deep Neural Networks for Rapid Chest CT-Based COVID-19 Patient Risk Assessment"

_ijerph, 2021, doi:10.3390/ijerph18062842_

Round 1
Reviewer 1 Report
The paper presents a new labeled and annotated CT dataset from Covid-19 patients, and a data-driven CT- Covid-19 patient risk assessment tool for lung segmentation and classification.
Comments:
- There is a mismatch between the title and the abstract. There is no mention of “Volume-of-interest aware deep neural networks” in the abstract.
- The introduction section lacks a discussion on the motivation of research. What knowledge gap are you trying to gap?
- Clearly state the novelty of your approach at the end of the Introduction section. What is your innovation?
- Discussion of related work should more focus on the methods specifically applied for COVID-19 CXR image analysis, segmentation and classification. Many recent successful methods are not mentioned. Also, discuss hybrid methods in which deep learning methods are combined with nature inspired optimization algorithms such as “COVID-19 image classification using deep features and fractional-order marine predators algorithm”. Some new and closely related papers are missed in references. It is important to cite and discuss the most important works in this area of research such as “A novel framework for rapid diagnosis of COVID-19 on computed tomography scans”. Pattern Analysis and Applications” (2021). Present a summary of related work as a table.
- 141: avoid mass-citing such as “[16,29,31,32,36–38]” but use each reference to support a separate statement or claim.
- The improvement of the proposed method over other models is rather small. Statistical analysis of the results must be performed. Calculate 95% confidence limits of the performance measures using the results from different cross-validation folds. Is the improvement statistically significant?
- Table in Figure 4: move to a separate numbered Table. Since it is average values from 320 slices, add standard deviation values or 95% confidence limits.
- Figure 7: what measure (metric) is represented by the colorbar?
- In the discussion section, discuss the limitations of the proposed model and threats-to-validity of the experimental results.
- Improve conclusions. Use the main numerical results to support your claims. Discuss future works and research perspectives.
Author Response
There is a mismatch between the title and the abstract. There is no mention of “Volume-of-interest aware deep neural networks” in the abstract.
Indeed, the abstract refers to Volume-of-Interest but does not include explicitly the term “Volume-of-interest aware deep neural networks”. The abstract has been updated accordingly.
The introduction section lacks a discussion on the motivation of research. What knowledge gap are you trying to gap?
The work reported in this paper is documenting a proposed approach for fast, AI-assisted, COVID patient detection and the criticality of the disease. Moreover, it presents the collected and annotated dataset from a reference hospital in Milan Italy during the first wave of COVID-19 in Europe. We have updated the introduction accordingly.
Clearly state the novelty of your approach at the end of the Introduction section. What is your innovation?
The innovation and contributions of our approach are documented at the end of the introduction section. To clarify them better, especially the Volume-of-Interest technique for more effective deep model training, this part has been updated in the manuscript by:
“....
- The publication and use of a new CT dataset of Covid-19 infected patients from Milan, a region early and intensively involved in the pandemic. The CTs are labeled according the patient hospitalization outcome.
- The introduction of a novel data-driven technique for spatially coherent sub-sampling of CT volumes by detecting Volumes-of-Interest (VoI). This leads to lightweight but highly informative CT data that allow us to train state-of-the-art 2D and 3D classifiers more effectively than existing techniques.
- The exhaustive experimentation among various CT-based risk-prediction approaches and 2D/3D classifiers for Covid-19 patient stratification, to assess their performance.
…”
Discussion of related work should more focus on the methods specifically applied for COVID-19 CXR image analysis, segmentation and classification. Many recent successful methods are not mentioned. Also, discuss hybrid methods in which deep learning methods are combined with nature inspired optimization algorithms such as “COVID-19 image classification using deep features and fractional-order marine predators algorithm”. Some new and closely related papers are missed in references. It is important to cite and discuss the most important works in this area of research such as “A novel framework for rapid diagnosis of COVID-19 on computed tomography scans”. Pattern Analysis and Applications” (2021). Present a summary of related work as a table.
The related work has been updated according to the comment of the reviewer which we also believe that helped to make this section more clear to the reader. On top of that, recent SoA works have been included as proposed “......Another very recent approach proposed in \cite{akram2021novel}, involves more traditional methods comprising feature extraction and selection via an entropy-based fitness optimizer, feature fusion to finally classify CT images as COVID-19 and healthy by utilising a Naive Bayes Classifier. Additionally, in \cite{chao2021integrative}, the authors employ Random Forests to classify if a patient needs ICU admission via a holistic approach, i.e. by employing CT as well as demographic and other related metadata for training........”
@article{akram2021novel,
title={A novel framework for rapid diagnosis of COVID-19 on computed tomography scans},
author={Akram, Tallha and Attique, Muhammad and Gul, Salma and Shahzad, Aamir and Altaf, Muhammad and Naqvi, S Syed Rameez and Dama{\v{s}}evi{\v{c}}ius, Robertas and Maskeli{\=u}nas, Rytis},
journal={Pattern analysis and applications},
pages={1--14},
year={2021},
publisher={Springer}
}
@article{chao2021integrative,
title={Integrative analysis for COVID-19 patient outcome prediction},
author={Chao, Hanqing and Fang, Xi and Zhang, Jiajin and Homayounieh, Fatemeh and Arru, Chiara D and Digumarthy, Subba R and Babaei, Rosa and Mobin, Hadi K and Mohseni, Iman and Saba, Luca and others},
journal={Medical Image Analysis},
volume={67},
pages={101844},
year={2021},
publisher={Elsevier}
}
141: avoid mass-citing such as “[16,29,31,32,36–38]” but use each reference to support a separate statement or claim.
As mentioned in the previous comment, the related work sections have been updated also including the comment.
The improvement of the proposed method over other models is rather small. Statistical analysis of the results must be performed. Calculate 95% confidence limits of the performance measures using the results from different cross-validation folds. Is the improvement statistically significant?
Following the reviewer's comment, we support our initial results with a null hypothesis test reported in line 419 of the revised manuscript and below. Particularly, our aim is to prove the claim that the performance of our best 2d model is better than the performance of the best 3d model as well from the radiologists. To that end, we can frame our hypothesis as follows:
- Ho: p1=p2 ; null hypothesis that states that two of the classifiers have equal performance
- Ha: p1<p2 ; alternative hypothesis that states that the second classifier have better performance
For accepting or rejecting our frame hypothesis we follow the test procedure process described in {Devore, Jay. Probability and Statistics for Engineering and the Sciences. Nelson Education, 2011.} and we calculate the test statistic value (Z) and define a test rejection region for each of the compared classifiers. The null hypothesis will then be rejected if and only if the computed test statistic value falls in the rejection region, while the lower-tailed test will prove our alternative hypothesis.
Table in Figure 4: move to a separate numbered Table. Since it is average values from 320 slices, add standard deviation values or 95% confidence limits.
Following the suggestion, we have separated the table from the figure and we have also included the standard deviation values of the results.
Figure 7: what measure (metric) is represented by the colorbar?
The class-discriminative localization maps (Figure.7) use the feature maps produced by the last convolutional layer of our CNN to assign a score for a specific class. Red regions correspond to a high score for the given class while blue corresponds to a low score. For simplicity, we normalize the scores in the interval [0,1], where 1 corresponds to a high score whilst 0 corresponds to a low score. The manuscript has been updated accordingly.
In the discussion section, discuss the limitations of the proposed model and threats-to-validity of the experimental results.
The discussion section has been updated including information about the limitations of the models and threats-to-validity of the experimental results due to the relatively limited dataset in comparison with the number of cases appear every day. A reference to the deployment and use in pilot phase of the models in HUMANITAS hospital will be the main monitoring tools to assess the potential weaknesses.
Improve conclusions. Use the main numerical results to support your claims. Discuss future works and research perspectives.
The conclusion section has been improved by discussing the numerical results and their positive impact on the medical experts that are interested in the deployment and use of the models in the hospital in the pilot phase. Further discussion with respect to finding and future work has been added, especially highlighting the lower performance of the 3D models in comparison with the 2D ones.

Reviewer 2 Report
The paper is very interesting as an application with CT lung images using deep learning techniques in a real dataset, which the authors are making available.
However, the text lacks many details and a deeper discussion. For example, important training parameters details are omitted, the sample size is small with a holdout validation, statistical comparation is absent and the multiclass measures are not well defined.
In this sense, I suggest the following modifications to improve the reading and the comprehension of the paper:
line 25
Please insert some brief statistics over covid-19 pandemic around the world, not just in Italy regions.
line 94
the term "deep models" is used without background of "deep learning". I advise improve this discussion around it in the section one as well as that every data driven model is a statistical model essentially.
line 151
I totally disagree with the term "old school" ML classifiers.
In fact, SVM - for example - is a more recent method than Neural Networks (deep or not). There are several papers which show that SVM (global solution) performs better than a deep learning method (local solution - sometimes). I advise cite some covid-19 image analysis using SVM and/or random forest, not just a general base reference.
line 165
exclude "otherwise black-box"
line 195
Please, add some explanation about the classes: moderate rick, severe risk and extreme risk. A more deep health overview.
line 275
I advise put the implementations procedure as a normal section. These details are too important to be a Appendix A. Also, please add in this section some details about the choice of learning rate, measures of convergence rate, number of parameters, convolutional layers, etc.
line 299
Please add a reference about oversampling replicates.
It's could be interesting to the readers.
line 338
Which kind of multi-class accuracy,
sensitivity, specificity, AUC, and F1-score? There are at least two of them: micro and macro measures.
Table 2-5
Please add time of training process.
A single 80/20 replication is not sufficient to compare predictive performance over many methods. Please, consider more replications or insert some of comparison statistical test
Table 4.
Why MixedConv-VoI is in bold?
There are VoI models with poor results, for example, F1-score with 20. Why? Add an explation in text.
In line 450
Please add a deeper comment about lower performance 3D models. Why?
Author Response
line 25
Please insert some brief statistics over covid-19 pandemic around the world, not just in Italy regions.
Indeed, we focused in Italy due to the fact that this study is based on data and work done in the HUMANITAS hospital, however it is essential to report the global picture. We updated the manuscript giving the current picture of the pandemic as reported by WHO.
“...According to World Health Organization, from February 2020 till now, 110 million Covid-19 cases have been confirmed while the world has suffered approximately 2.5 million losses. …”
line 94
the term "deep models" is used without background of "deep learning". I advise improve this discussion around it in the section one as well as that every data driven model is a statistical model essentially.
We have updated the manuscript to avoid this mismatch. On top of that, we have updated the introduction section in order to better bind the importance of the introduced dataset and the proposed statistical data-driven models. The added text in the manuscript can be found below:
“...To this end, the work reported in this paper is documenting a new data-driven approach for fast, AI-assisted, Covid patient detection and the criticality of the disease. Moreover, it presents COVID-19\textunderscore CHDSET Dataset, a newly collected and annotated dataset from a reference hospital in Milan Italy during the first wave of Covid-19 in Europe. To this end, we exploit the data contained to our new Covid-19 CT annotated dataset and propose a novel CT-based stratification approach to rapidly assess the infection severity and then, the risk for Covid-19 patients. The statistical variety of the data collected from patients with varying characteristics (as discussed in Sec. \ref{sec:dataset}), allows us to train our models more effectively due to the relationship between a dependent variable and a set of independent variables (statistical models)...”
line 151
I totally disagree with the term "old school" ML classifiers.
In fact, SVM - for example - is a more recent method than Neural Networks (deep or not). There are several papers which show that SVM (global solution) performs better than a deep learning method (local solution - sometimes). I advise cite some covid-19 image analysis using SVM and/or random forest, not just a general base reference.
Indeed, the term is totally inaccurate and it has been removed from the manuscript. Furthermore, we included recent Covid-19 methods that use these classifiers but also keeping the original general references since in this part of the manuscript, we already discuss recent methods that use them for Covid-19 patient classification.
line 165
exclude "otherwise black-box"
Removed.
line 195
Please, add some explanation about the classes: moderate rick, severe risk and extreme risk. A more deep health overview.
The class labeling was conducted exclusively from the actual hospitalization path each patient underwent, meaning that there is no “clinical” pattern for each risk level since patients are assessed based on lung infection rate, age, clinical factors and other comorbidities. This information has been added in the manuscript as a footnote to better inform the reader for the labeling procedure.
line 275
I advise put the implementations procedure as a normal section. These details are too important to be a Appendix A. Also, please add in this section some details about the choice of learning rate, measures of convergence rate, number of parameters, convolutional layers, etc.
Following the reviewer’s suggestion, we moved the implementation details of our various deep models in the manuscript and specifically at the end of the results section per task. On top of that, we updated text to include further details such as the model parameters. The learning rates were selected as the rates that were able to drive the models to convergence with the highest pace.
line 299
Please add a reference about oversampling replicates.
It's could be interesting to the readers.
Indeed it would be beneficial for the reader, so we inserted in the manuscript the reference for the oversampling technique we followed.
@book{brownlee2020imbalanced,
title={Imbalanced classification with Python: better metrics, balance skewed classes, cost-sensitive learning},
author={Brownlee, Jason},
year={2020},
publisher={Machine Learning Mastery}
}
line 338
Which kind of multi-class accuracy,
sensitivity, specificity, AUC, and F1-score? There are at least two of them: micro and macro measures.
Indeed we omitted to describe it explicitly. We refer to macro-measures so we have updated the manuscript accordingly.
Table 2-5
Please add time of training process.
The training time per epoch was approximately 40’-50’ for the 2D and 50’-60’ for the 3D classifiers with the oversampled dataset. The duration is estimated roughly by the logs without using specific metrics so we prefer not including this information in the manuscript.
A single 80/20 replication is not sufficient to compare predictive performance over many methods. Please, consider more replications or insert some of comparison statistical test
Following the reviewer's comment, we support our initial results with a null hypothesis test reported in line 419 of the revised manuscript and below. Particularly, our aim is to prove the claim that the performance of our best 2d model is better than the performance of the best 3d model as well from the radiologists. To that end, we can frame our hypothesis as follows:
- Ho: p1=p2 ; null hypothesis that states that two of the classifiers have equal performance
- Ha: p1<p2 ; alternative hypothesis that states that the second classifier have better performance
For accepting or rejecting our frame hypothesis we follow the test procedure process described in {Devore, Jay. Probability and Statistics for Engineering and the Sciences. Nelson Education, 2011.} and we calculate the test statistic value (Z) and define a test rejection region for each of the compared classifiers. The null hypothesis will then be rejected if and only if the computed test statistic value falls in the rejection region, while the lower-tailed test will prove our alternative hypothesis.
Table 4.
Why MixedConv-VoI is in bold?
All VoI model names were in bold in order to be highlighted in comparison with the models based on other strategies. To avoid confusion, we removed the bold style from the model names from all tables.
There are VoI models with poor results, for example, F1-score with 20. Why? Add an explation in text.
Given that there is no VoI model with F1-score equal to 20% in Table 4 (new table 5), we assume that there was a confusion with RVoI which is for random VoI selection.
In line 450
Please add a deeper comment about lower performance 3D models. Why?
Indeed, this was not expected given the capability of the 3D models to learn spatially coherent features due to the 3D input. This constitutes one of the current tasks for further research in our lab, thus we included it in the discussion and future steps.

Reviewer 3 Report
In this paper, the authors design and evaluate a data-driven CT-based tool for automatic Covid-19 patient risk assessment (discharged, hospitalized, intensive care unit) based on lung infection quantization through segmentation and, subsequently, CT classification. Authors tackle the high and varying dimensionality of the CT input by detecting and analyzing only a sub-volume of the CT, the Volume-of-Interest (VoI), and proves the effectiveness of our VoI-based approach. The authors have carried out a very meaningful study, which can provide some help for the diagnosis and treatment of covid-19. However, there are still some shortcomings.
(1) In this paper, the introduction of deep learning method is not particularly sufficient, which can not clearly see what innovative work the authors have done on the basis of existing methods.
(2) At present, the number of people infected with Covid-19 in the world has reached more than 100 million, while the data set tested in this paper is only a few hundred, which is a little small, and the validation of the method is not particularly sufficient.
Author Response
(1) In this paper, the introduction of deep learning method is not particularly sufficient, which can not clearly see what innovative work the authors have done on the basis of existing methods.
We updated the introduction section of the manuscript according to the reviewer’s comment in a way that gives a better explanation of the proposed method. In particular, the updated part can be found below:
“.....To this end, the work reported in this paper is documenting a new data-driven approach for fast, AI-assisted, Covid patient detection and the criticality of the disease. Moreover, it presents COVID-19\textunderscore CHDSET Dataset, a newly collected and annotated dataset from a reference hospital in Milan Italy during the first wave of Covid-19 in Europe. To this end, we exploit the data contained to our new Covid-19 CT annotated dataset and propose a novel CT-based stratification approach to rapidly assess the infection severity and then, the risk for Covid-19 patients. The statistical variety of the data collected from patients with varying characteristics (as discussed in Sec. \ref{sec:dataset}), allows us to train our models more effectively due to the relationship between a dependent variable and a set of independent variables (statistical models). In detail, we introduce a two-stage data-driven approach to classify Covid-19 patients into three risk classes (\textit{moderate, severe, extreme}), based on their risk to be, respectively, discharged, hospitalize, or sent to ICU short after CT examination. We firstly train a pixel-wise segmentation model on lung CT slices to detect the volume-of-interest (VoI), i.e. the sub-volume of the lung containing the most infected slices, instead of the whole CT volume, which contains healthy or non-informative slices and whose elevated voxel-grid resolution would require an abrupt down-sampling. Secondly, based on the detected VoI, we train various 2D and 3D classifiers to predict the Covid-19 patient risks. This two-stage cascade concept (data of interest selection → classification) is not new in the medical imaging field \cite{liu2018deep, liu2019fully, mei2020artificial}; however, to the best of our knowledge, our approach is the first that is driven by a spatially coherent data representation that leads to better and robust performance.
Summarizing, the main contributions of the present research work are reported in the following:
- The publication and use of COVID-19\textunderscore CHDSET Dataset, a new CT dataset of Covid-19 infected patients from Milan, a region early and intensively involved in the pandemic. The CTs are labeled according the patient hospitalization outcome.
- The introduction of a novel, staged, data-driven technique of spatially coherent sub-sampling of CT volumes by detecting Volumes-of-Interest (VoI). This leads to lightweight but highly informative CT data that allow us to train state-of-the-art 2D and 3D classifiers more effectively than existing techniques.
- The exhaustive experimentation among various CT-based risk-prediction approaches and 2D/3D classifiers for Covid-19 patient stratification, to assess their performance. …”
(2) At present, the number of people infected with Covid-19 in the world has reached more than 100 million, while the data set tested in this paper is only a few hundred, which is a little small, and the validation of the method is not particularly sufficient.
Indeed the infected cases are already more than 100 millions and all over the world that makes every case quite unique and probably will guide to failures for many data driven models that are biased to data with poor variety in characteristics. On the other hand, public COVID CT datasets with pixel-wise annotations are limited both from the number and the capacity perspective (https://arxiv.org/abs/2003.13865, https://www.medrxiv.org/content/10.1101/2020.04.24.20078584v3) and that is due to the difficulty to allocate resources (experienced MDs-radiologists) to carefully identify the anomalies and annotate the lesions. Exactly for that reason we believe that the publication of this dataset constitutes a small in comparison with the number of Covid-19 cases, but significant contribution to the community, where more teams are going to contribute more data in order to further enrich the data pool.

Reviewer 4 Report
Authors designed and evaluated a data-driven CT-based tool for automatic Covid-19 patient risk assessment (discharged, hospitalized, intensive care unit) based on lung infection quantization through segmentation
and, subsequently, CT classification. They introduced a new, two-stage method to assess the severeness of Covid-19 patients’ exclusively based on chest CT volumes.
Authors concluded that data-driven models achieve better performance when processing consecutive slices centered on the mostly infected sub-volume of the CT 3D models. Furthermore, that providing one-shot decision on VoI, show lower performance when compared to models decomposing the 3D data into 2D per-slice 455 predictions aggregated with majority voting.
Authors benchmarked their approach against other strategies. The paper is well organized, written and referred. It is an important and necessary result for current global situation.
Author Response
Thank you for your effort to review and evaluate our manuscript.

Round 2
Reviewer 1 Report
The authors have revised the article very well, addressing all my comments. I recommend the article to be published.
Reviewer 3 Report
accept